# Node-layer duality in networked systems

Charley Presigny[1], Marie-Constance Corsi[1] & Fabrizio De Vico Fallani [1]✉

Real-world networks typically exhibit several aspects, or layers, of interactions among their nodes. By permuting the role of the nodes and the layers, we establish a new criterion to construct the dual of a network. This approach allows to examine connectivity from either a node-centric or layer-centric viewpoint. Through rigorous analytical methods and extensive simulations, we demonstrate that nodewise and layerwise connectivity measure different but related aspects of the same system. Leveraging node-layer duality provides complementary insights, enabling a deeper comprehension of diverse networks across social science, technology and biology. Taken together, these findings reveal previously unappreciated features of complex systems and provide a fresh tool for delving into their structure and dynamics.

Duality belongs to noticeable concepts of philosophy, social, and natural systems that refer to different, often antithetic, aspects of the same phenomenon or entity. Duality enhances our understanding of complex systems by highlighting complementary properties with important implications in theoretical studies and real-world applications. For example, in quantum physics elementary particles such as electrons can behave as both discrete elements and continuous waves. In electrical engineering, voltage-current duality allows to transform a circuit problem into an analogous one that may be easier to solve or analyze.

Complex systems are often characterized by many local interactions giving rise to emerging properties that affect the structure and dynamics of the global network[1–4]. Apart from a few related efforts[5–7], the question of whether complex networks exhibit non-trivial forms of duality has remained poorly explored. This is partly because in the classical formalism where the nodes are the sole interacting units, it is hard to identify genuinely distinct aspects of the system[8]. Recent developments in multilayer network theory offer a unique opportunity to overcome this limitation.

In a multilayer network, nodes are connected through multiple types of interactions or relationships (layers), creating a more comprehensive representation of the system[9–13]. Real-world examples include transportation networks where commuters travel via different modalities and social networks where individuals exchanging information via different technological media[14], as well as brain networks where neurons interact on different anatomical and functional scales[15]. Hence, in real-world scenarios the nodes constitute the entities of the system and the layers correspond to traits or aspects of those entities.

Many quantities from classical networks have been extended to multilayer networks to unveil non-trivial properties based on how the entities (i.e., nodes) interact[16–28]. Interestingly, complementary properties could be obtained by looking at how their different aspects (i.e., the layers) are interconnected, too. While recent works have started exploiting layerwise properties to characterize the system clustering[29,30], ranking[31–33] and redundancy[34], the relationship with the nodewise counterpart is still unclear as well as their complementary role. To fill in this critical gap, we propose a general framework that exploits the intrinsic properties of multilayer networks and offers a dual characterization of the same system.

Multilayer networks are mathematically represented by supra-adjacency matrices whose entry $X_{ij}^{\alpha\beta}$ contains the connectivity between the node $i$ at layer $\alpha$ and the node $j$ at layer $\beta$[11,14,28]. Here, the basic interacting unit is not just the node, but the so-called node-layer duplet[26]. By swapping the indices of the nodes and the layers, we define the dual network $P^{\mathsf{T}}XP = Y$ where $P$ is an opportune permutation matrix[35,36]. Since this is a symmetric transformation it is always possible to go back to the primal version $X = PYP^{\mathsf{T}}$. Put simply, networks can be represented as nodes connected through layers or equivalently as layers connected through nodes (Fig. 1). At the global level, the information captured by the two representations is the same because node-layer duplets are representation-invariant, i.e., $X_{ij}^{\alpha\beta} = Y_{\alpha\beta}^{ij}$.

Locally, the information captured by the main units in the primal and dual representation is in general different. For example, let us consider the multidegree centrality as a simple intuitive measure of local connectivity[37]. In the primal nodewise representation, here referred as to $\mathcal{X}$, the nodes are the main units of interest and integrate information across the layers. The multidegree centrality of node $i$

[1]Sorbonne Université, Institut du Cerveau - Paris Brain Institute - ICM, CNRS, Inria, Inserm, AP-HP, Hôpital de la Pitié Salpêtrière, Paris, France.
✉e-mail: fabrizio.de-vico-fallani@inria.fr

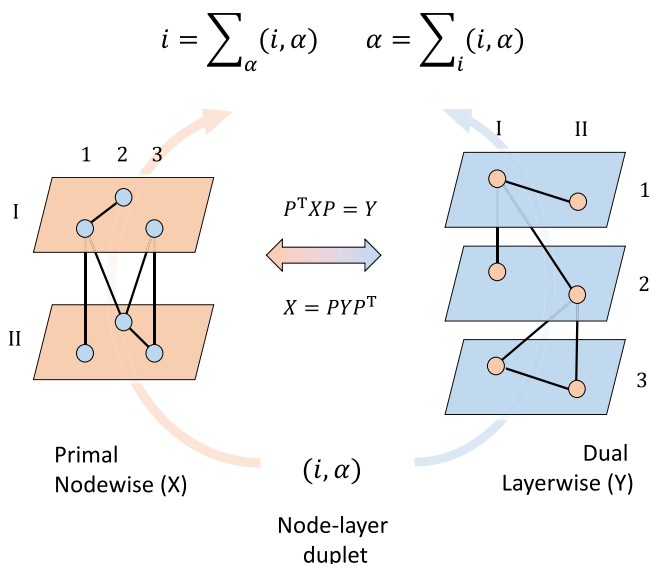

$$i = \sum_{\alpha}(i,\alpha) \qquad \alpha = \sum_{i}(i,\alpha)$$

$P^{T}XP = Y$

$X = PYP^{T}$

Primal
Nodewise ($X$)

$(i,\alpha)$

Node-layer
duplet

Dual
Layerwise ($Y$)

**Fig. 1 | Conceptual model of duality in complex networks.** Real networks typically consist of entities (nodes) interacting across different modes or aspects (layers). The unitary element of such multilayer networks is the node-layer duplet. In the left side illustration, a duplet identifies a node $i = 1, 2, 3$ and a layer $\alpha = I, II$. By opportunely permuting the indices ($P^{T}XP = Y$) nodes become layers (blue) and layers become nodes (red) without altering the local connectivity (right side). By construction, replica links become intralayer, intralayer links become replica, and interlayer links stay interlayer. This transformation is symmetric and it is always possible to go back to the primal version $X = PYP^{T}$. The same system can be, therefore, equivalently represented as entities connected through aspects or aspects connected through entities. Two complementary descriptions can be then obtained depending on the representation side. In the primal nodewise (left), connectivity is integrated across aspects and one looks at how entities are interacting. In the dual layerwise (right), connectivity is integrated across entities and one rather looks instead at how their aspects are interconnected.

reads $k_{\mathcal{X}}^{i} = \sum_{j\alpha\beta} X_{ij}^{\alpha\beta}$. In the dual layerwise representation $\mathcal{Y}$, the layers become the main units and integrate information across the nodes. The multidegree centrality of layer $\alpha$ is $k_{\mathcal{Y}}^{\alpha} = \sum_{ij\beta} Y_{\alpha\beta}^{ij}$. In the following, we demonstrate how alterations to the connections within the system are differently perceived by the multidegree centralities in each representation.

## Results

### Node-layer duality captures complementary network changes

To assess the impact of edge perturbations on both nodewise and layerwise representations, we introduced a model that randomly rewires an arbitrary network, with the type of link rearrangement determined by three parameters. Specifically, the likelihood of rewiring a link while *i)* keeping the layers unchanged ($p_{node}$), *ii)* keeping the nodes unchanged ($p_{layer}$), and *iii)* altering both the nodes and layers, akin to a "teleportation" process ($p_{tel}$) **(Methods)**. The total probability associated with the rewiring event satisfies $p_{node} + p_{layer} + p_{tel} = 1$ (Fig. 2a).

By using a time-discrete Markov chain approach[38] we first demonstrated that the expected value for the multidegree centrality, i.e., the sum of all the links connected to a node *i* (or a layer $\alpha$), after randomly rewiring a percentage *r* of links, can be explicitly derived from the initial multidegree centrality sequence. For sufficiently sparse and large multilayer networks, a compact form reads:

$$k_{\mathcal{X}}^{i}(r) = k_{\mathcal{X}}^{i} + r\left(\mu_{\mathcal{X}} - k_{\mathcal{X}}^{i}\right)(1 - p_{layer})$$
$$k_{\mathcal{Y}}^{\alpha}(r) = k_{\mathcal{Y}}^{\alpha} + r\left(\mu_{\mathcal{Y}} - k_{\mathcal{Y}}^{\alpha}\right)(1 - p_{node}) \tag{1}$$

where $\mu$ stands for the average multidegree centrality (Text S1.4−5). Individual multidegree centralities increase or decrease linearly with the amount of rewiring depending on whether their value is lower or higher than the average. The magnitude of local change is further modulated by the type of rewiring, i.e., $1 - p_{layer} = p_{node} + p_{tel}$ for nodewise and $1 - p_{node} = p_{layer} + p_{tel}$ for layerwise (Supplementary Fig. S1).

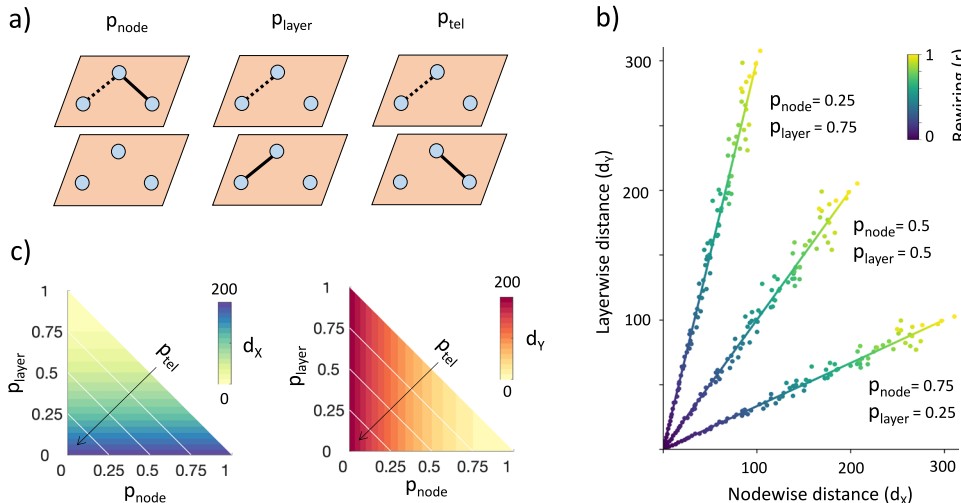

**Fig. 2 | Complementary properties of nodewise and layerwise connectivity. a** Schematic illustration of the different types of edge rewiring. The type of perturbation is determined by the probability to make a link displacement while *i)* keeping the layers unaltered ($p_{node}$), *ii)* keeping the nodes unaltered ($p_{layer}$), *iii)* altering both nodes and layers ($p_{tel}$). Dotted lines = old position, solid lines = new position. **b** Linear relation between layerwise and nodewise global connectivity changes. Global changes are computed as Euclidean distances ($d$) between multidegree centrality vectors. Lower slopes (higher $d_{\mathcal{X}}$) are obtained for $p_{node} > p_{layer}$. Higher slopes (higher $d_{\mathcal{Y}}$) are obtained for $p_{layer} > p_{node}$. Solid lines correspond to

the theoretical formulas in the case of random networks with $N = M = 200$, connection density $q = 0.0005$, for the entire rewiring range *r* (color line). Scattered points correspond to synthetic random networks simulated with the same parameters. **c** Global connectivity changes as a function of rewiring parameters. Nodewise distances ($d_{\mathcal{X}}$) increase linearly with $p_{node}$ (x-axis) and $p_{tel}$ (white diagonals) but they cannot see edge displacements that keep nodes unvaried ($p_{layer} \to 1$). Layerwise distances ($d_{\mathcal{Y}}$) increase linearly with $p_{layer}$ (y-axis) and $p_{tel}$ (white diagonals) but they are blind to edge displacements that keep layers unvaried ($p_{node} \to 1$).

Using Eq. (1), we analytically derived the expression of the global connectivity change as the Euclidean distance of the multidegree centrality vectors $k_\chi$ and $k_y$ before and after rewiring:

$$d_\chi(r) = r\sqrt{N}\sigma_\chi(1 - p_{layer})$$
$$d_y(r) = r\sqrt{M}\sigma_y(1 - p_{node}), \quad (2)$$

where $\sigma$ denotes standard deviation of the multidegree centrality (Text S1.5, and Supplementary Fig. S2). In general, both distances increase with the system size, the heterogeneity of the multidegree centrality, as well as with the type and amount of rewiring. Specifically, $d_\chi$ increases with the probability that the rewiring alters the connected nodes ($p_{layer} \to 0$), while $d_y$ increases with the probability that the rewiring alters the connected layers ($p_{node} \to 0$).

From Eq. (2), we also showed that the layerwise distances scale linearly with nodewise distances by a factor that depends only on the intrinsic characteristics of the system $c = \frac{\sqrt{M}\sigma_y}{\sqrt{N}\sigma_\chi}$ and on the type of rewiring (Fig. 2b, and Text S1.7):

$$d_y(r) = c\frac{1 - p_{node}}{1 - p_{layer}}d_\chi(r) \propto d_\chi(r) \quad (3)$$

Notably, perturbations that involve different nodes with layers being unaltered ($p_{node} = 1$) are not captured by layerwise distances. Similarly, perturbations involving different layers with nodes kept unvaried ($p_{layer} = 1$) cannot be captured by nodewise distances (Fig. 2c).

The constant $c$ requires the knowledge of the multidegree centrality standard deviations. In the case of random networks, where links are initially distributed in a random fashion across nodes and layers, standard deviations can be analytically derived in the limit of large systems and it is trivial to prove that $c = 1$ (Text S1.7). More complex expressions can also be derived in the case of finite-size networks exhibiting power-law multidegree centrality distributions (Text S1.8). In practice, for many real-world networks standard deviations can always be calculated from the system under investigation.

Taken together, Eqs. (2) and (3) demonstrate the complementarity of nodewise and layerwise connectivity and establish a general form of duality in complex networks based on the structural invariance of node-layer permutation.

## Dual classification of real-world multiplex networks

Since real-world multilayer networks have very different numbers of nodes and layers, we first studied the effects of different system sizes on nodewise and layerwise distances. To do so, we considered complete uniform rewirings starting from random multilayer and multiplex networks, the latter exhibiting interlayer connections solely between replica nodes. In this situation, both $p_{node}$ and $p_{layer}$ vanish with the network size leading to $d_y \simeq d_\chi \propto NM$ for multilayers and $d_y \propto \sqrt{2MN} > d_\chi \propto \sqrt{MN}$ for multiplexes (Texts S1.7, S1.9).

This means that a very different number of layers and nodes give in general lower distances in random systems. Instead, the largest distances occur when $N = M$ in multilayer configurations and $N = M + \frac{N+M}{3} > M$ in multiplex ones. Notably in random multiplexes $d_Y \propto \sqrt{2}d_\chi$ by construction, so that their dual layerwise representation a-priori better emphasizes differences with respect to the primal nodewise (Fig. 3, and Supplementary Fig. S3).

Next, we evaluated the ability of the node-layer duality to provide a better separation of different real multiplex systems, from transportation and social networks to genetic and neuronal interactions[23,24,34,39–43]. Here, we used the explicit formulas to obtain the nodewise and layerwise distances of the real networks from the completely uniform rewired counterparts. This avoided to perform heavy numerical computations for very big systems (e.g., Twitter $N = 88804$, $M = 3$). Finally, to get rid of possible different network size effects, we

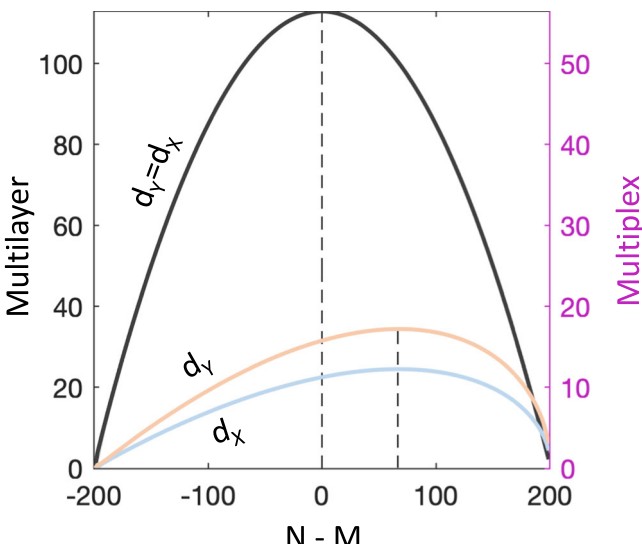

**Fig. 3 | Different size effects for multilayer and multiplex configurations.** In multilayer random networks, the largest global connectivity changes obtained after uniform rewiring occur in both nodewise and layerwise representations when the number of nodes $N$ equal the number of layers $M$ (black lines). However, in multiplex random networks the maximum change, as measured by the Euclidean distance ($d$) between multidegree centrality vectors, is reached when there are more nodes than layers (i.e., $N = M + (N+M)/3$, Text S1.10). In this plot $N$ and $M$ vary in a way to ensure the condition $N + M = 200$ so that $N = M + 200/3$. In addition, layerwise distances (orange) are by construction higher than nodewise distances (blue). These findings suggest that layerwise representations might be a-priori better candidates to spout out topological differences in multiplex networks, as compared to standard nodewise (Text S1.7). Solid lines correspond to the theoretical formulas for random networks with connection density $q = 0.0005$, rewiring ratio $r = 0.5$, and uniform rewiring probability (Text S1.9).

normalized the actual distances by those obtained from equivalently rewired random multiplex networks (Texts S2, S1.4, S1.7, and S1.9).

Results confirm that adding the layerwise representation allows for a better separation of multiplex networks, which would be otherwise indistinguishable by only looking at nodewise distances (Fig. 4a). Such separation is mainly due to the higher heterogeneity of the layer multidegree centralities as compared to nodewise representation (Tab S1). This can be visually appreciated by the different distributions in the projection plots showing how nodes contribute to layers, and viceversa (Fig. 4b, **Methods**).

Using a $k$-means clustering algorithm we found that networks tend to optimally separate into two subgroups (Supplementary Fig. S4). Those with relatively low $d_\chi$ and $d_y$ values mostly correspond to systems spatially embedded (e.g., German transport, EUAir, C.Elegans connectome). Instead, those with relatively high distances lack of strong spatial connotations (e.g., PierreAuger, Arxiv, Twitter events). While there are few exceptions (e.g., Human microbiome), these results reflect the typical limited node degree heterogeneity of spatial networks due to environmental physical constraints[44].

## Latent brain frequency reorganization in Alzheimer's disease

We used the node-layer duality framework to identify abnormal network signatures of Alzheimer's disease (AD) across different brain areas and frequencies of brain activity. To do so, we considered full multilayer brain networks estimated from source-reconstructed magnetoencephalography (MEG) signals in a group of 23 AD human subjects matched with a group of 27 healthy controls HC[45,46] (**Methods**). Specifically, we used bispectral coherence to simultaneously infer weighted interactions between regions of interest (the nodes) and between signal frequencies (the layers)[47].

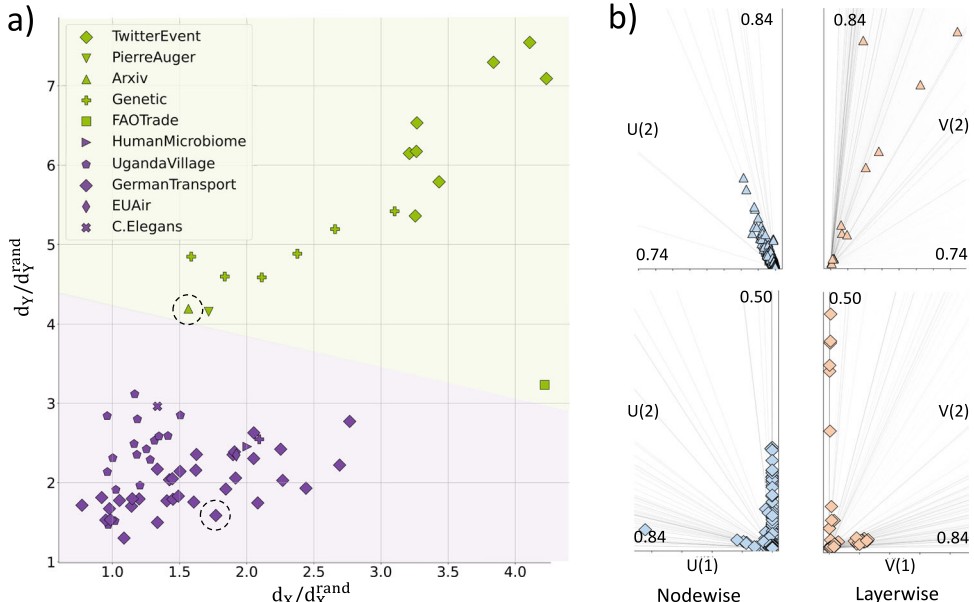

**Fig. 4 | Dual characterization of real-world multiplex networks. a** Log-log scatter plot of the nodewise ($x$-axis) and layerwise ($y$-axis) connectivity distances from uniformly rewired counterparts (Text S1.8). To avoid network-size biases, all values are further divided by the distances obtained rewiring equivalent random networks. The layer representation enables a better classification of networks that would be otherwise indistinguishable (e.g., Arxiv versus German transport highlighted by dashed circles). Networks optimally group into two clusters almost perfectly matching the spatial (violet) and non-spatial (green) nature of the systems ($k$-means=2, Silhouette score = 0.70, Supplementary Fig. S4). **b** Projection plots of two representative multiplex networks (i.e., German transport and Arxiv). In the nodewise, markers correspond to nodes and gray lines to layers. In the layerwise, markers correspond to layers and gray lines to nodes (Methods). Differently from Arxiv (top), the markers in the German network (bottom) tend to accumulate on few main lines meaning that both nodes and layers tend to contribute preferentially to few components. Also, values in the layerwise tend to span larger intervals in comparison with nodewise, indicating the presence of more heterogenous multi-degree centrality distributions in the layerwise representation (Text S2). The standard deviation of the Arxiv's layer multidegree centrality ($\sigma_Y = 8931$) is significantly higher than the German transport ($\sigma_Y = 80.39$), and this is eventually reflected by the relative higher layerwise distance $d_Y$ in panel a).

Results indicate that AD is characterized by functional network disruptions in both nodewise and layerwise representations. In the nodewise, the brain regions implicated in the atrophy process exhibit a loss of multidegree centrality, here computed as the sum of all the weighted links connected to a node. In the layerwise, several brain frequencies within the *alpha* range (8–13 Hz) in the AD group present reduced multidegree centralities when compared to HC (Fig. 5a, b).

While these local connectivity differences are not statistically significant, we find that globally frequency-wise distances can better discriminate AD patients as compared to region-wise distances. In particular, the coarser the frequency resolution, the more significant the difference between $d_y$ and $d_x$ (Fig. 5c). This behavior does not solely result from the reduction in network size, but instead originates from the greater consistency in connectivity changes within frequencies compared to between frequencies (Supplementary Fig. S5).

Finally, we found that both node and layer multidegree centrality decrements are significantly associated with more severe cognitive decline of AD patients as measured by the mini-mental state examination (MMSE)[48]. In the brain space, the most predictive area is the bilateral caudal anterior cingulate cortex, a well-known hub of information processing in the brain that plays an essential role in AD pathophysiology[49,50] (Supplementary Table S2). Notably, a higher number of stronger correlations emerges in the frequency space. The most significant ones lie within the *alpha* frequency range, considered one of the most reliable noninvasive functional predictors of AD-related cognitive symptoms[51,52] (Fig. 5d, and Supplementary Table S3).

## Discussion

Whether networks express duality, a general concept characterized by the coexistence and interplay of complementary aspects, is poorly understood. Drawing inspiration from recent advances in multilayer network theory, we propose a rigorous framework that reveals dual manifestations of the same system.

We show that edge perturbations that do not alter the local connectivity of the nodes (i.e., $p_{layer} = 1$) are only visible in the dual layerwise representation. Conversely, changes that do not alter the local connectivity of the layers (i.e., $p_{node} = 1$) are only visible in the primal nodewise representation. This exclusive duality can be relaxed by tuning the type of rewiring thus yielding partial information about nodewise and layerwise aspects simultaneously. For a given $p_{tel}$, the more information a particular instance gives about one, the less it will give about the other (Fig. 2b). The continuous complementarity that emerges is not unique to node-layer duality but can also be observed in other contexts, such as the wave-particle duality relation[53,54].

In general, the amount of measured information increases with the variability of the local connectivity Eq. (2). Compared to full multilayer configurations, the variability of the layerwise connectivity in multiplex random networks is by construction higher than the nodewise counterpart. This implies that layerwise representations may be more suitable for studying real-world systems for which interlayer connections are frequently lacking. Node-layer duality is not only important from a fundamental perspective but also because it would allow us to describe difficult models in terms of their simpler dual counterparts or improve problem-solving efficiency by equating primal and dual expansions. Furthermore, while these results are obtained using relatively simple network quantities and models, other features can be explored within this framework to enrich the overall characterization and result interpretation.

To demonstrate the significance of our approach, we examine the brain, which can be represented as a multilayer network spanning various scales and levels[15]. In particular, the interaction between neural oscillations at different frequencies allows for coordinating and integrating information across different brain regions and cognitive

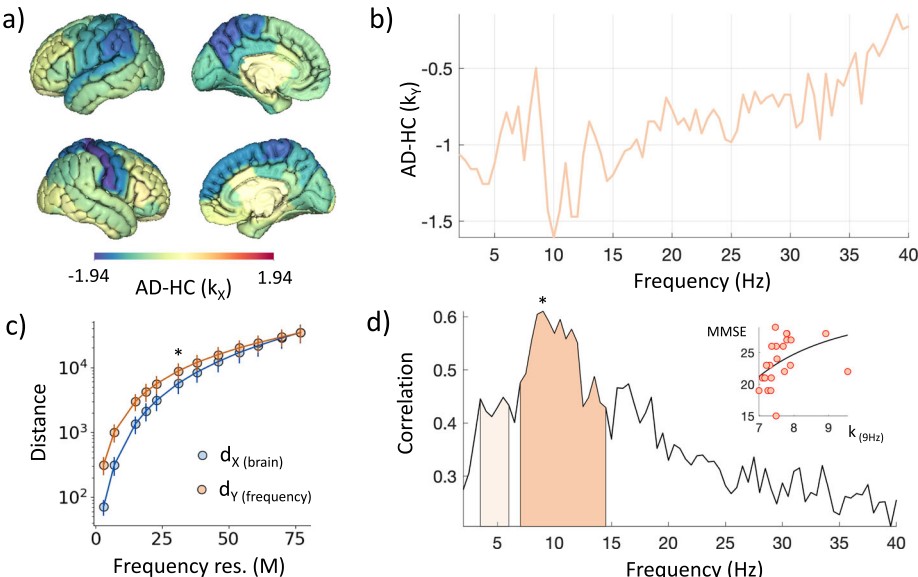

**Fig. 5 | Alzheimer's disease multilayer brain network disruption in region- and frequency-wise representations.** Multilayer brain networks are inferred from source-reconstructed MEG signals using cross-frequency coupling. In the primal representation, nodes correspond to brain regions ($N = 70$) and layers to different frequency bins ($M = 77$). Both intralayer and interlayer links are provided, estimating the amount of activity interaction (Methods). **a** Statistical difference (Wilcoxon test, Z-score) between brain region multidegree centralities of AD patients and healthy controls (HC). **b** Statistiscal difference (Wilcoxon test, Z-score) between brain frequency multidegree centralities of AD patients and healthy controls (HC). **c** Group-averaged nodewise and layerwise distances between AD and HC for different frequency resolutions (from $M = 77$ to $M = 3$). The asterisk marks the number of layers ($M \leq 31$) for which distances become significantly different ($p < 0.05$, FDR corrected). Vertical bars denote standard deviations. **d** Spearman correlation is computed between the frequency multidegree centralities of AD patients and their cognitive decline scores (MMSE). Colored areas show the significant ranges obtained from a cluster-based permutation procedure for multiple correlations[63]. Darker color $p = 0.0374$; lighter color $p = 0.049$. The inset spots out the AD patients' MMSE as a function of the multidegree centrality at $9\,Hz$, giving the highest significant correlation $R = 0.601$ as indicated by the asterisk. Regressing curves resulting from a square fit are shown for illustrative purposes.

processes[55]. Understanding disruptions in cross-frequency coupling is therefore crucial to identifying biomarkers and novel treatments in neurological disorders, such as Alzheimer's disease[56]. The node-layer duality reveals that the cognitive decline of AD patients is better predicted by dual changes among frequencies of brain activity rather than primal changes among regions. Although increasing the sample size, enriching the network information, and using alternative methods can improve the overall prediction in general[57,58], our approach offers the first proof-of-concept accounting for node-layer duality in brain networks. More in general, node-layer duality translates to the ability to uncover fresh insights into modern neuroscience, such as understanding how neurons communicate via parallel frequency channels[59] and elucidating how local brain damages lead to structure-function reorganization[60]. These findings can significantly improve our models of brain functioning during cognitive or motor tasks and contribute to the identification of predictive biomarkers for brain diseases, such as neurodegeneration and stroke recovery.

We hope that the concept of node-layer duality will stimulate fresh investigations of complex interconnected systems, with broad implications in a wide range of disciplines, including network science, systems biology, and social network analysis.

## Methods
### Stochastic rewiring network model
Starting with a network of $N$ nodes, $M$ layers and $L$ links arbitrarily distributed, the algorithm first randomly selects a proportion $r \leq L$ of edges to rewire. Then, for each selected edge, a new position is randomly drawn based on the relative importance of the rewiring parameters $p_{node}$, $p_{layer}$, and $p_{tel}$, which determine in a probabilistic fashion the type of edge reassignment. For example, $p_{node} = 0.3$, $p_{layer} = 0.6$ and $p_{tel} = 0.1$ implies that about 30% of the selected edges will be rewired without altering the initial connected layers, 60% without altering the nodes, and 10% by altering both layers and nodes. During the process,

if a new position is already occupied or if it alters the original nature of the network (e.g., multiplex or multilayer), another position is proposed following the same probabilistic rule. Note that when $p_{node} = 1$, the layer multidegree centrality sequence is entirely preserved. Similarly, when $p_{layer} = 1$, the node multidegree centrality sequence is conserved. More details in Text S1.1–4.

### Projection plots for multilayer networks
To help understand how the nodes contribute to layers and vice versa, we adapted the method proposed by[61] to visualize partitioned networks. To do so, we first compute the contribution matrix $C$ containing the number of links that a node $i$ shares with a layer $\alpha$. We next consider a typical truncated singular value decomposition (tSVD) $C = U\Sigma V^{\dagger}$, where $\Sigma$ is a diagonal matrix containing the singular values, and $U$ and $V$ are respectively the left and right orthogonal matrices associated with the nodewise and layerwise representation. To visualize the nodewise contribution we project the space spanned by the first two left singular vectors, i.e., $U(1)$-$U(2)$. In this 2D space, the layers are represented as lines whose direction depends on their cohesiveness, i.e., layers that share many links tend to be represented along similar directions. The nodes are instead represented as points. The more the nodes contribute to a specific layer the more they tend to be aligned to its direction. The distance of each point from the origin is finally proportional to its multidegree centrality. Similar visualization can be obtained for the layerwise contributions by projecting the space spanned by the first two right singular vectors, i.e., $V(1)$-$V(2)$.

### Multilayer brain network construction
Multilayer brain networks are obtained from the experimental data published in[46]. We refer to this paper for more detailed descriptions. 23 Alzheimer's disease (AD) patients and 27 healthy age-matched control (HC) subjects, participated in the study. All participants underwent the Mini-Mental State Examination (MMSE) for global cognition[48]. For each

subject, 6 minutes resting-state eyes-closed brain activity was recorded noninvasively using a whole-head MEG system with 102 magnetometers and 204 planar gradiometers (Elekta Neuromag TRIUX MEG system) at a sampling rate of 1000 Hz. Signal artifacts were removed using different techniques, including signal space separation, principal component analysis, and visual inspection. Finally, source-imaging was used to project the signals from the sensor to the source space consisting of $N = 70$ regions of interest (ROI) defined by the Lausanne cortical atlas parcellation[62]. Here, we used spectral bicoherence to estimate functional connectivity between ROIs and between frequencies of brain activity[47]. Specifically, we considered $M = 77$ layers corresponding to frequencies in the 2−40 Hz range with a resolution of 0.5 Hz. Other parameters were non-overlapping windows of 2s averaged according to the Welch method. We finally symmetrized the resulting supra-adjacency matrices by selecting the highest value of bicoherence between each pair of ROIs and frequencies. The resulting networks are *full-multilayer* consisting of both intralayer and interlayer connections, including weighted links between replica nodes.

## Data availability

The brain network data generated in this study have been deposited in the Zenondo database and freely accessible and usable under the Creative Common license BY 4.0 license (https://doi.org/10.5281/zenodo.12099874).

## Code availability

All the code used to generate the results is freely available, documented and usable under the MIT license (https://github.com/Inria-NERV/multilayer_duality).

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

## Acknowledgements

FDVF acknowledges support from the European Research Council (ERC) under the European Union Horizon 2020 research and innovation program (Grant Agreement No. 864729).

## Author contributions

C.P. conceived the study, performed theoretical modeling and analysis, wrote the paper, and prepared the figures; M.C. performed data processing for the brain network construction; and F.D.V.F. conceived and coordinated the study, wrote the paper, and prepared the figures.

## Competing interests

The authors declare that they have no conflict of interest and that the content is solely the responsibility of the authors and does not necessarily represent the official views of any of the funding agencies.
