## [Peer Review file · Nature Communications]

Node-layer duality in networked systems

Corresponding Author: Professor Fabrizio De Vico Fallani

This manuscript has been previously reviewed at another journal. This document only contains reviewer comments, rebuttal and decision letters for versions considered at Nature Communications.

Version 0:

Reviewer comments:

Reviewer #1

(Remarks to the Author)

I have carefully read the rebuttal letter and the revised manuscript. I believe the authors have done a remarkable work in amending the article and I take here the occasion to thank them for having considered constructively the concerns and criticism I raised with my previous reports on this piece. I believe the authors have toned down the narrative around their concept of "duality" and have now reached a balanced presentation of their results. The broad interest of the authors' framework is now much more visible as well as its potentials for future research. The application to brain data makes the manuscript perfectly aligned with the interdisciplinary interests of the readership of Nature Communications and I do hope this work will receive the interdisciplinary attention it deserves. I am therefore happy to recommend publication and I congratulate the authors for their result.
